# The Importance of Communities for Learning to Influence

**Eric Balkanski**
Harvard University
ericbalkanski@g.harvard.edu

**Nicole Immorlica**
Microsoft Research
nicimm@microsoft.com

**Yaron Singer**
Harvard University
yaron@seas.harvard.edu

## Abstract

We consider the canonical problem of influence maximization in social networks. Since the seminal work of Kempe, Kleinberg, and Tardos [KKT03] there have been two, largely disjoint efforts on this problem. The first studies the problem associated with learning the generative model that produces cascades, and the second focuses on the algorithmic challenge of identifying a set of influencers, assuming the generative model is known. Recent results on learning and optimization imply that in general, if the generative model is not known but rather learned from training data, no algorithm for influence maximization can yield a constant factor approximation guarantee using polynomially-many samples, drawn from any distribution.

In this paper we describe a simple algorithm for maximizing influence from training data. The main idea behind the algorithm is to leverage the strong community structure of social networks and identify a set of individuals who are influentials but whose communities have little overlap. Although in general, the approximation guarantee of such an algorithm is unbounded, we show that this algorithm performs well experimentally. To analyze its performance, we prove this algorithm obtains a constant factor approximation guarantee on graphs generated through the stochastic block model, traditionally used to model networks with community structure.

## 1 Introduction

For well over a decade now, there has been extensive work on the canonical problem of influence maximization in social networks. First posed by Domingos and Richardson [DR01, RD02] and elegantly formulated and further developed by Kempe, Kleinberg, and Tardos [KKT03], influence maximization is the algorithmic challenge of selecting individuals who can serve as early adopters of a new idea, product, or technology in a manner that will trigger a large cascade in the social network.

In their seminal paper, Kempe, Kleinberg, and Tardos characterize a family of natural influence processes for which selecting a set of individuals that maximize the resulting cascade reduces to maximizing a submodular function under a cardinality constraint. Since submodular functions can be maximized within a $1 - 1/e$ approximation guarantee, one can then obtain desirable guarantees for the influence maximization problem. There have since been two, largely separate, agendas of research on the problem. The first line of work is concerned with learning the underlying submodular function from observations of cascades [LK03, AA05, LMF+07, GBL10, CKL11, GBS11, NS12, GLK12, DSSY12, ACKP13, DSGRZ13, FK14, DBB+14, CAD+14, DGSS14, DLBS14, NPS15, HO15]. The second line of work focuses on algorithmic challenges revolving around maximizing influence,

assuming the underlying function that generates the diffusion process is known [KKT05, MR07, SS13, BBCL14, HS15, HK16, AS16].

In this paper, we consider the problem of learning to influence where the goal is to maximize influence from observations of cascades. This problem synthesizes both problems of learning the function from training data and of maximizing influence given the influence function. A natural approach for learning to influence is to first learn the influence function from cascades, and then apply a submodular optimization algorithm on the function learned from data. Somewhat counter-intuitively, it turns out that this approach yields desirable guarantees only under very strong learnability conditions[1]. In some cases, when there are sufficiently many samples, and one can observe exactly which node attempts to influence whom at every time step, these learnability conditions can be met. A slight relaxation however (e.g. when there are only partial observations [NPS15, HXKL16]), can lead to sharp inapproximability.

A recent line of work shows that even when a function is statistically learnable, optimizing the function learned from data can be inapproximable [BRS17, BS17]. In particular, even when the submodular function $f : 2^N \to \mathbb{R}$ is a coverage function (which is PMAC learnable [BDF$^+$12, FK14]), one would need to observe exponentially many samples $\{S_i, f(S_i)\}_{i=1}^m$ to obtain a constant factor approximation guarantee. Since coverage functions are special cases of the well studied models of influence (independent cascade, linear and submodular threshold), this implies that when the influence function is not known but learned from data, the influence maximization problem is intractable.

**Learning to influence social networks.** As with all impossibility results, the inapproximability discussed above holds for worst case instances, and it may be possible that such instances are rare for influence in social networks. In recent work, it was shown that when a submodular function has bounded curvature, there is a simple algorithm that can maximize the function under a cardinality constraint from samples [BRS16]. Unfortunately, simple examples show that submodular functions that dictate influence processes in social networks do not have bounded curvature. Are there other reasonable conditions on social networks that yield desirable approximation guarantees?

**Main result.** In this paper we present a simple algorithm for learning to influence. This algorithm leverages the idea that social networks exhibit strong community structure. At a high level, the algorithm observes cascades and aims to select a set of nodes that are influential, but belong to different communities. Intuitively, when an influential node from a certain community is selected to initiate a cascade, the marginal contribution of adding another node from that same community is small, since the nodes in that community were likely already influenced. This observation can be translated into a simple algorithm which performs very well in practice. Analytically, since community structure is often modeled using stochastic block models, we prove that the algorithm obtains a constant factor approximation guarantee in such models, under mild assumptions.

## 1.1 Technical overview

The analysis for the approximation guarantees lies at the intersection of combinatorial optimization and random graph theory. We formalize the intuition that the algorithm leverages the community structure of social networks in the standard model to analyze communities, which is the stochastic block model. Intuitively, the algorithm obtains good approximations by picking the nodes that have the largest individual influence while avoiding picking multiple nodes in the same community by pruning nodes with high influence overlap. The individual influence of nodes and their overlap are estimated by the algorithm with what we call first and second order marginal contributions of nodes, which can be estimated from samples. We then uses phase transition results of Erdős–Rényi random graphs and branching processes techniques to compare these individual influences for nodes in different communities in the stochastic block model and bound the overlap of pairs of nodes.

**The optimization from samples model.** Optimization from samples was recently introduced by [BRS17] in the context of submodular optimization, we give the definition for general set functions.

**Definition 1.** *A class of functions $\mathcal{F} = \{f : 2^N \to \mathbb{R}\}$ is $\alpha$-**optimizable from samples over distribution** $\mathcal{D}$ under constraint $\mathcal{M}$ if there exists an algorithm s.t. for all $f \in \mathcal{F}$, given a set of samples $\{(S_i, f(S_i))\}_{i=1}^m$ where the sets $S_i$ are drawn i.i.d. from $\mathcal{D}$, the algorithm returns $S \in \mathcal{M}$ s.t.:*

$$\Pr_{S_1,\ldots,S_m \sim \mathcal{D}} \left[ \mathbb{E}[f(S)] \geq \alpha \cdot \max_{T \in \mathcal{M}} f(T) \right] \geq 1 - \delta,$$

*where the expectation is over the decisions of the algorithm and $m \in \mathrm{poly}(|N|, 1/\delta)$.*

We focus on bounded product distributions $\mathcal{D}$, so every node $a$ is, independently, in $S \sim \mathcal{D}$ with some probability $p_a \in [1/\mathrm{poly}(n), 1 - 1/\mathrm{poly}(n)]$. We assume this is the case throughout the paper.

**Influence process.** We assume that the influence process follows the standard *independent cascade* model. In the independent cascade model, a node $a$ influences each of its neighbors $b$ with some probability $q_{ab}$, independently. Thus, given a seed set of nodes $S$, the set of nodes influenced is the number of nodes connected to some node in $S$ in the random subgraph of the network which contains every edge $ab$ independently with probability $q_{ab}$. We define $f(S)$ to be the expected number of nodes influenced by $S$ according to the independent cascade model over some weighted social network.

**The learning to influence model: optimization from samples for influence maximization.** The learning to influence model is an interpretation of the optimization from samples model [BRS17] for the specific problem of influence maximization in social networks. We are given a collection of samples $\{(S_i, |cc(S_i)|)\}_{i=1}^m$ where sets $S_i$ are the seed sets of nodes and $|cc(S_i)|$ is the number of nodes influenced by $S_i$, i.e., the number of nodes that are connected to $S_i$ in the random subgraph of the network. This number of nodes is a random variable with expected value $f(S_i) := \mathbb{E}[|cc(S_i)|]$ over the realization of the influence process. Each sample is an independent realization of the influence process. The goal is then to find a set of nodes $S$ under a cardinality constraint $k$ which maximizes the influence in expectation, i.e., find a set $S$ of size at most $k$ which maximizes the expected number of nodes $f(S)$ influenced by seed set $S$.

## 2 The Algorithm

We present the main algorithm, COPS. This algorithm is based on a novel optimization from samples technique which detects overlap in the marginal contributions of two different nodes, which is useful to avoid picking two nodes who have intersecting influence over a same collection of nodes.

### 2.1 Description of COPS

COPS, consists of two steps. It first orders nodes in decreasing order of first order marginal contribution, which is the expected marginal contribution of a node $a$ to a random set $S \sim \mathcal{D}$. Then, it iteratively removes nodes $a$ whose marginal contribution overlaps with the marginal contribution of at least one node before $a$ in the ordering. The solution is the $k$ first nodes in the pruned ordering.

---

**Algorithm 1** COPS, learns to influence networks with COmmunity Pruning from Samples.

---

**Input:** Samples $\mathcal{S} = \{(S, f(S))\}$, acceptable overlap $\alpha$.

    Order nodes according to their first order marginal contributions

    Iteratively remove from this ordering nodes $a$ whose marginal contribution has overlap of at least $\alpha$ with at least one node before $a$ in this ordering.

    **return** $k$ first nodes in the ordering

---

The strong performance of this algorithm for the problem of influence maximization is best explained with the concept of communities. Intuitively, this algorithm first orders nodes in decreasing order of their individual influence and then removes nodes which are in a same community. This second step allows the algorithm to obtain a diverse solution which influences multiple different communities of the social network. In comparison, previous algorithms in optimization from samples [BRS16, BRS17] only use first order marginal contributions and perform well if the function is close to linear. Due to the high overlap in influence between nodes in a same community, influence functions are far

from being linear and these algorithms have poor performance for influence maximization since they only pick nodes from a very small number of communities.

## 2.2 Computing overlap using second order marginal contributions

We define second order marginal contributions, which are used to compute the overlap between the marginal contribution of two nodes.

**Definition 2.** *The* second order expected marginal contribution *of a node $a$ to a random set $S$ containing node $b$ is*

$$v_b(a) := \mathop{\mathbb{E}}_{S \sim \mathcal{D}: a \notin S, b \in S} [f(S \cup \{a\}) - f(S)].$$

The first order marginal contribution $v(a)$ of node $a$ is defined similarly as the marginal contribution of a node $a$ to a random set $S$, i.e., $v(a) := \mathbb{E}_{S \sim \mathcal{D}: a \notin S}[f(S \cup \{a\}) - f(S)]$. These contributions can be estimated arbitrarily well for product distributions $\mathcal{D}$ by taking the difference between the average value of samples containing $a$ and $b$ and the average value of samples containing $b$ but not $a$ (see Appendix B for details).

The subroutine $\text{OVERLAP}(a, b, \alpha)$, $\alpha \in [0, 1]$, compares the second order marginal contribution of $a$ to a random set containing $b$ and the first order marginal contribution of $a$ to a random set. If $b$ causes the marginal contribution of $a$ to decrease by at least a factor of $1 - \alpha$, then we say that $a$ has marginal contribution with overlap of at least $\alpha$ with node $b$.

---

**Algorithm 2** $\text{OVERLAP}(a, b, \alpha)$, returns true if $a$ and $b$ have marginal contributions that overlap by at least a factor $\alpha$.

---

**Input:** Samples $\mathcal{S} = \{(S, f(S))\}$, node $a$, acceptable overlap $\alpha$

If second order marginal contribution $v_b(a)$ is at least a factor of $1 - \alpha$ smaller than first order marginal contribution $v(a)$,

**return** Node $a$ has overlap of at least $\alpha$ with node $b$

---

$\text{OVERLAP}$ is used to detect nodes in a same community. In the extreme case where two nodes $a$ and $b$ are in a community $C$ where any node in $C$ influences all of community $C$, then the second order marginal contribution $v_b(a)$ of $a$ to random set $S$ containing $b$ is $v_b(a) = 0$ since $b$ already influences all of $C$ so $a$ does not add any value, while $v(a) \approx |C|$. In the opposite case where $a$ and $b$ are in two communities which are not connected in the network, we have $v(a) = v_b(a)$ since adding $b$ to a random set $S$ has no impact on the value added by $a$.

## 2.3 Analyzing community structure

The main benefit from COPS is that it leverages the community structure of social networks. To formalize this explanation, we analyze our algorithm in the standard model used to study the community structure of networks, the stochastic block model. In this model, a fixed set of nodes $V$ is partitioned in communities $C_1, \ldots, C_\ell$. The network is then a random graph $G = (V, E)$ where edges are added to $E$ independently and where an intra-community edge is in $E$ with much larger probability than an inter-community edge. These edges are added with identical probability $q_C^{\text{sb}}$ for every edge in a same community, but with different probabilities for edges inside different communities $C_i$ and $C_j$. We illustrate this model in Figure 1.

# 3 Dense Communities and Small Seed Set in the Stochastic Block Model

In this section, we show that COPS achieves a $1 - O(|C_k|^{-1})$ approximation, where $C_k$ is the $k$th largest community, in the regime with dense communities and small seed set, which is described below. We show that the algorithm picks a node from each of the $k$ largest communities with high probability, which is the optimal solution. In the next section, we show a constant factor approximation algorithm for a generalization of this setting, which requires a more intricate analysis.

In order to focus on the main characteristics of the community structure as an explanation for the performance of the algorithm, we make the following simplifying assumptions for the analysis. We

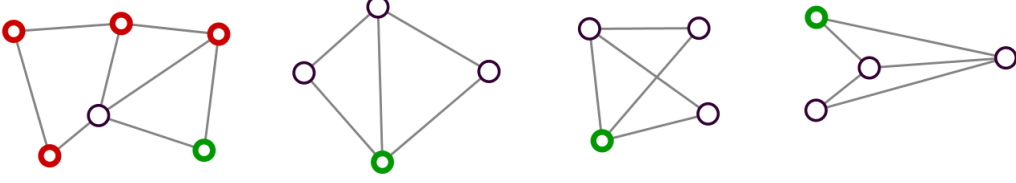

Figure 1: An illustration of the stochastic block model with communities $C_1$, $C_2$, $C_3$ and $C_4$ of sizes $6, 4, 4$ and $4$. The optimal solution for influence maximization with $k = 4$ is in green. Picking the $k$ first nodes in the ordering by marginal contributions without pruning, as in [BRS16], leads to a solution with nodes from only $C_1$ (red). By removing nodes with overlapping marginal contributions, COPS obtains a diverse solution.

first assume that there are no inter-community edges.[2] We also assume that the random graph obtained from the stochastic block model is redrawn for every sample and that we aim to find a good solution in expectation over both the stochastic block model and the independent cascade model.

Formally, let $G = (V, E)$ be the random graph over $n$ nodes obtained from an independent cascade process over the graph generated by the stochastic block model. Similarly as for the stochastic block model, edge probabilities for the independent cascade model may vary between different communities and are identical within a single community $C$, where all edges have weights $q_C^{\text{ic}}$. Thus, an edge $e$ between two nodes in a community $C$ is in $E$ with probability $p_C := q_C^{\text{ic}} \cdot q_C^{\text{sb}}$, independently for every edge, where $q_C^{\text{ic}}$ and $q_C^{\text{sb}}$ are the edge probabilities in the independent cascade model and the stochastic block model respectively. The total influence by seed set $S$ is then $|\text{cc}_G(S_i)|$ where $\text{cc}_G(S)$ is the set of nodes connected to $S$ in $G$ and we drop the subscript when it is clear from context. Thus, the objective function is $f(S) := \mathbb{E}_G[|\text{cc}(S)|]$. We describe the two assumptions for this section.

**Dense communities.** We assume that for the $k$ largest communities $C$, $p_C > 3 \log |C|/|C|$ and $C$ has super-constant size ($|C| = \omega(1)$). This assumption corresponds to communities where the probability $p_C$ that a node $a_i \in C$ influences another node $a_j \in C$ is large. Since the subgraph $G[C]$ of $G$ induced by a community $C$ is an Erdős–Rényi random graph, we get that $G[C]$ is connected with high probability (see Appendix C).

**Lemma 3.** *[ER60] Assume $C$ is a "dense" community, then the subgraph $G[C]$ of $G$ is connected with probability $1 - O(|C|^{-2})$.*

**Small seed set.** We also assume that the seed sets $S \sim \mathcal{D}$ are small enough so that they rarely intersect with a fixed community $C$, i.e., $\Pr_{S \sim \mathcal{D}}[S \cap C = \emptyset] \geq 1 - o(1)$. This assumption corresponds to cases where the set of early influencers is small, which is usually the case in cascades.

The analysis in this section relies on two main lemmas. We first show that the first order marginal contribution of a node is approximately the size of the community it belongs to (Lemma 4). Thus, the ordering by marginal contributions orders elements by the size of the community they belong to. Then, we show that any node $a \in C$ that is s.t. that there is a node $b \in C$ before $a$ in the ordering is pruned (Lemma 5). Regarding the distribution $S \sim \mathcal{D}$ generating the samples, as previously mentioned, we consider any bounded product distribution. This implies that w.p. $1 - 1/\text{poly}(n)$, the algorithm can compute marginal contribution estimates $\tilde{v}$ that are all a $1/\text{poly}(n)$-additive approximation to the true marginal contributions $v$ (See Appendix B for formal analysis of estimates). Thus, we give the analysis for the true marginal contributions, which, with probability $1 - 1/\text{poly}(n)$ over the samples, easily extends for arbitrarily good estimates.

The following lemma shows that the ordering by first order marginal contributions corresponds to the ordering by decreasing order of community sizes that nodes belong to.

**Lemma 4.** *For all $a \in C$ where $C$ is one of the $k$ largest communities, the first order marginal contribution of node $a$ is approximately the size of its community, i.e., $(1 - o(1))|C| \leq v(a) \leq |C|$.*

*Proof.* Assume $a$ is a node in one of the $k$ largest communities. Let $\mathcal{D}_a$ and $\mathcal{D}_{-a}$ denote the distributions $S \sim \mathcal{D}$ conditioned on $a \in S$ and $a \notin S$ respectively. We also denote marginal contributions by $f_S(a) := f(S \cup \{a\}) - f(S)$. We obtain

$$v(a) = \underset{S \sim \mathcal{D}_{-a}, G}{\mathbb{E}}[f_S(a)] \geq \underset{S \sim \mathcal{D}_{-a}}{\Pr}[S \cap C = \emptyset] \cdot \underset{G}{\Pr}[\mathrm{cc}(a) = C] \cdot \underset{\substack{S \sim \mathcal{D}_{-a} : S \cap C = \emptyset, \\ G : \mathrm{cc}(a) = C}}{\mathbb{E}}[f_S(a)]$$

$$= \underset{S \sim \mathcal{D}_{-a}}{\Pr}[S \cap C = \emptyset] \cdot \underset{G}{\Pr}[\mathrm{cc}(a) = C] \cdot |C|$$

$$\geq (1 - o(1)) \cdot |C|$$

where the last inequality is by the small seed set assumption and since $C$ is connected with probability $1 - o(1)$ (Lemma 3 and $|C| = \omega(1)$ by dense community assumption). For the upper bound, $v(a)$ is trivially at most the size of $a$'s community since there are no inter-community edges. $\qquad\square$

The next lemma shows that the algorithm does not pick two nodes in a same community.

**Lemma 5.** *With probability $1 - o(1)$, for all pairs of nodes $a, b$ such that $a, b \in C$ where $C$ is one of the $k$ largest communities,* $\mathrm{OVERLAP}(a, b, \alpha) = $ *True for any constant $\alpha \in [0, 1)$.*

*Proof.* Let $a, b$ be two nodes in one of the $k$ largest communities $C$ and $\mathcal{D}_{-a,b}$ denote the distribution $S \sim \mathcal{D}$ conditioned on $a \notin S$ and $b \in S$. Then,

$$v_b(a) = \underset{S \sim \mathcal{D}_{-a,b}}{\mathbb{E}}[f_S(a)] \leq \Pr[b \in \mathrm{cc}(a)] \cdot 0 + \Pr[b \notin \mathrm{cc}(a)] \cdot |C| = o(1) \leq o(1) \cdot v(a)$$

where the last equality is since $G[C]$ is not connected w.p. $O(|C|^{-2})$ by Lemma 3 and since $|C| = \omega(1)$ by the dense community assumption, which concludes the proof. $\qquad\square$

By combining Lemmas 4 and 5, we obtain the main result for this section (proof in Appendix D).

**Theorem 6.** *In the dense communities and small seed set setting,* COPS *with $\alpha$-overlap allowed, for any constant $\alpha \in (0, 1)$ is a $1 - o(1)$-approximation algorithm for learning to influence from samples from a bounded product distribution $\mathcal{D}$.*

## 4 Constant Approximation for General Stochastic Block Model

In this section, we relax assumptions from the previous section and show that COPS is a constant factor approximation algorithm in this more demanding setting. Recall that $G$ is the random graph obtained from both the stochastic block model and the independent cascade model. A main observation that is used in the analysis is to observe that the random subgraph $G[C]$, for some community $C$, is an Erdős–Rényi random graph $G_{|C|, p_C}$.

**Relaxation of the assumptions.** Instead of only considering dense communities where $p_C = \Omega((\log |C|)/|C|)$, we consider both *tight* communities $C$ where $p_C \geq (1 + \epsilon)/|C|$ for some constant $\epsilon > 0$ and *loose* communities $C$ where $p_C \leq (1 - \epsilon)/|C|$ for some constant $\epsilon > 0$.[3] We also relax the small seed set assumption to the reasonable *non-ubiquitous* seed set assumption. Instead of having a seed set $S \sim \mathcal{D}$ rarely intersect with a fixed community $C$, we only assume that $\Pr_{S \sim \mathcal{D}}[S \cap C = \emptyset] \geq \epsilon$ for some constant $\epsilon > 0$. Again, since seed sets are of small sizes in practice, it seems reasonable that with some constant probability a community does not contain any seeds.

**Overview of analysis.** At a high level, the analysis exploits the remarkably sharp threshold for the phase transition of Erdős–Rényi random graphs. This phase transition (Lemma 7) tells us that a tight community $C$ contains w.h.p. a giant connected component with a constant fraction of the nodes from $C$. Thus, a single node from a tight community influences a constant fraction of its community in expectation. The ordering by first order marginal contributions thus ensures a constant factor approximation of the value from nodes in tight communities (Lemma 10). On the other hand, we show that a node from a loose community influences only at most a constant number of nodes in expectation (Lemma 8) by using branching processes. Since the algorithm checks for overlap using second order marginal contributions, the algorithm picks at most one node from any tight community (Lemma 11). Combining all the pieces together, we obtain a constant factor approximation (Theorem 12).

We first state the result for the giant connected component in a tight community, which is an immediate corollary of the prominent giant connected component result in the Erdős–Rényi model.

**Lemma 7.** *[ER60] Let $C$ be a tight community with $|C| = \omega(1)$, then $G[C]$ has a "giant" connected component containing a constant fraction of the nodes in $C$ w.p. $1 - o(1)$.*

The following lemma analyzes the influence of a node in a loose community through the lenses of Galton-Watson branching processes to show that such a node influences at most a constant number of nodes in expectation. The proof is deferred to Appendix E.

**Lemma 8.** *Let $C$ be a loose community, then $f(\{a\}) \leq c$ for all $a \in C$ and some constant c.*

We can now upper bound the value of the optimal solution $S^\star$. Let $C_1, \ldots, C_t$ be the $t \leq k$ tight communities that have at least one node in $C_i$ that is in the optimal solution $S^\star$ and that are of super-constant size, i.e., $|C| = \omega(1)$. Without loss, we order these communities in decreasing order of their size $|C_i|$.

**Lemma 9.** *Let $S^\star$ be the optimal set of nodes and $C_i$ and $t$ be defined as above. There exists a constant c such that $f(S^\star) \leq \sum_{i=1}^{t} |C_i| + c \cdot k$.*

*Proof.* Let $S_A^\star$ and $S_B^\star$ be a partition of the optimal nodes in nodes that are in tight communities with super-constant individual influence and nodes that are not in such a community. The influence $f(S_A^\star)$ is trivially upper bounded by $\sum_{i=1}^{t} |C_i|$. Next, there exists some constant $c$ s.t. $f(S_B^\star) \leq \sum_{a \in S_B^\star} f(\{a\}) \leq c$· where the first inequality is by submodularity and the second since nodes in loose communities have constant individual influence by Lemma 8 and nodes in tight community without super-constant individual influence have constant influence by definition. We conclude that by submodularity, $f(S^\star) \leq f(S_A^\star) + f(S_B^\star) \leq \sum_{i=1}^{t} |C_i| + c \cdot k$. $\qquad\square$

Next, we argue that the solution returned by the algorithm is a constant factor away from $\sum_{i=1}^{t} |C_i|$.

**Lemma 10.** *Let $a$ be the $i$th node in the ordering by first order maginal contribution after the pruning and $C_i$ be the $i$th largest tight community with super-constant individual influence and with at least one node in the optimal solution $S^\star$. Then, $f(\{a\}) \geq \epsilon|C_i|$ for some constant $\epsilon > 0$.*

*Proof.* By definition of $C_i$, we have $|C_1| \geq \cdots \geq |C_i|$ that are all tight communities. Let $b$ be a node in $C_j$ for $j \in [i]$, $\mathbb{1}_{\mathrm{gc}(C)}$ be the indicator variable indicating if there is a giant component in community $C$, and $\mathrm{gc}(C)$ be this giant component. We get

$$v(b) \geq \Pr[\mathbb{1}_{\mathrm{gc}(C_j)}] \cdot \Pr_{S \sim \mathcal{D}_{-b}}[S \cap C_j = \emptyset] \cdot \Pr[b \in \mathrm{gc}(C_j)] \cdot \mathbb{E}[|\mathrm{gc}(C_j)| : b \in \mathrm{gc}(C_j)]$$

$$\geq (1 - o(1)) \cdot \epsilon_1 \cdot \epsilon_2 \cdot \epsilon_3|C_j| \geq \epsilon|C_j|$$

for some constants $\epsilon_1, \epsilon_2, \epsilon_3, \epsilon > 0$ by Lemma 7 and the non-ubiquitous assumption. Similarly as in Theorem 6, if $a$ and $b$ are in different communities, OVERLAP$(a, b, \alpha) = $ False for $\alpha \in (0, 1]$. Thus, there is at least one node $b \in \cup_{j=1}^{i} C_j$ at position $i$ or after in the ordering after the pruning, and $v(b) \geq \epsilon|C_j|$ for some $j \in [i]$. By the ordering by first order marginal contributions and since node $a$ is in $i$th position, $v(a) \geq v(b)$, and we get that $f(\{a\}) \geq v(a) \geq v(b) \geq \epsilon|C_j| \geq \epsilon|C_i|$. $\qquad\square$

Next, we show that the algorithm never picks two nodes from a same tight community and defer the proof to Appendix E.

**Lemma 11.** *If $a, b \in C$ and $C$ is a tight community, then OVERLAP$(a, b, \alpha) = $ True for $\alpha = o(1)$.*

We combine the above lemmas to obtain the approximation guarantee of COPS (proof in Appendix E).

**Theorem 12.** *With overlap allowed $\alpha = 1/\operatorname{poly}(n)$, COPS is a constant factor approximation algorithm for learning to influence from samples drawn from a bounded product distribution $\mathcal{D}$ in the setting with tight and loose communities and non-ubiquitous seed sets.*

## 5   Experiments

In this section, we compare the performance of COPS and three other algorithms on real and synthetic networks. We show that COPS performs well in practice, it outperforms the previous optimization from samples algorithm and gets closer to the solution obtained when given complete access to the influence function.

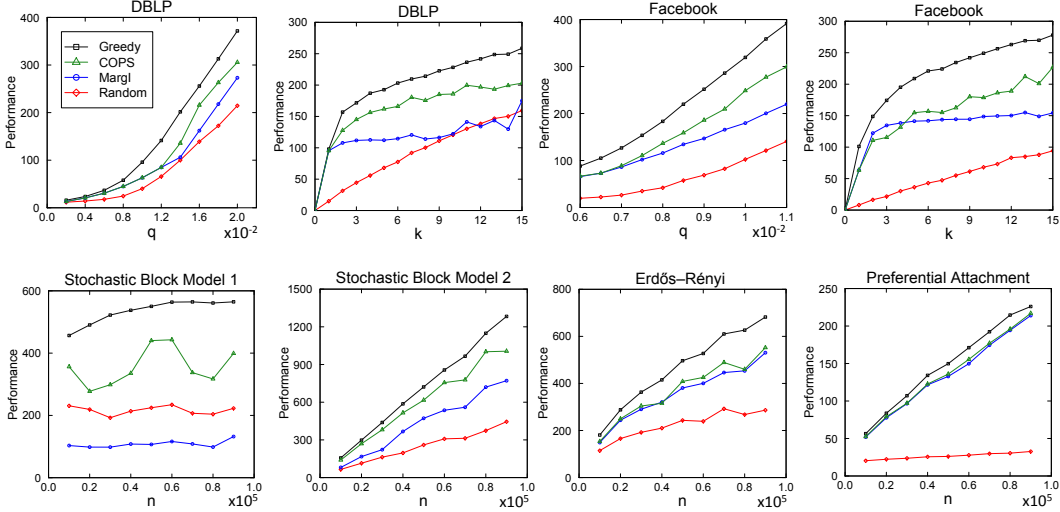

Figure 2: Empirical performance of COPS against the GREEDY upper bound, the previous optimization from samples algorithm MARGI and a random set.

**Experimental setup.** The first synthetic network considered is the stochastic block model, SBM 1, where communities have random sizes with one community of size significantly larger than the other communities. We maintained the same expected community size as $n$ varied. In the second stochastic block model, SBM 2, all communities have same expected size and the number of communities was fixed as $n$ varied. The third and fourth synthetic networks were an Erdős–Rényi (ER) random graph and the preferential attachment model (PA). Experiments were also conducted on two real networks publicly available ([LK15]). The first is a subgraph of the Facebook social network with $n = 4k$ and $m = 88k$. The second is a subgraph of the DBLP co-authorship network, which has ground truth communities as described in [LK15], where nodes of degree at most 10 were pruned to obtain $n = 54k$, $m = 361k$ and where the $1.2k$ nodes with degree at least 50 were considered as potential nodes in the solution.

**Benchmarks.** We considered three different benchmarks to compare the COPS algorithm against. The standard GREEDY algorithm in the value query model is an upper bound since it is the optimal efficient algorithm given value query access to the function and COPS is in the more restricted setting with only samples. MARGI is the optimization from samples algorithm which picks the $k$ nodes with highest first order marginal contribution ([BRS16]) and does not use second order marginal contributions. RANDOM simply returns a random set. All the samples are drawn from the product distribution with marginal probability $k/n$, so that samples have expected size $k$. We further describe the parameters of each plot in Appendix F.

**Empirical evaluation.** COPS significantly outperforms the previous optimization from samples algorithm MARGI, getting much closer to the GREEDY upper bound. We observe that the more there is a community structure in the network, the better the performance of COPS is compared to MARGI, e.g., SBM vs ER and PA (which do not have a community structure). When the edge weight $q := q^{\text{i.c.}}$ for the cascades is small, the function is near-linear and MARGI performs well, whereas when it is large, there is a lot of overlap and COPS performs better. The performance of COPS as a function of the overlap allowed (experiment in Appendix F) can be explained as follows: Its performance slowly increases as the the overlap allowed increases and COPS can pick from a larger collection of nodes until it drops when it allows too much overlap and picks mostly very close nodes from a same community. For SBM 1 with one larger community, MARGI is trapped into only picking nodes from that larger community and performs even less well than RANDOM. As $n$ increases, the number of nodes influenced increases roughly linearly for SBM 2 when the number of communities is fixed since the number of nodes per community increases linearly, which is not the case for SBM 1.

## Footnotes

[1]In general, the submodular function $f : 2^N \to \mathbb{R}$ needs to be learnable *everywhere* within arbitrary precision, i.e. for every set $S$ one needs to assume that the learner can produce a surrogate function $\tilde{f} : 2^N \to \mathbb{R}$ s.t. for every $S \subseteq N$ the surrogate guarantees to be $(1 - \epsilon)f(S) \le \tilde{f}(S) \le (1 + \epsilon)f(S)$, for $\epsilon \in o(1)$[HS16, HS17].

[2]The analysis easily extends to cases where inter-community edges form with probability significantly smaller to $q_C^{\text{sb}}$, for all $C$.

[3]Thus, we consider all possible sizes of communities except communities of size that converges to *exactly* $1/p_C$, which is unlikely to occur in practice.

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
