[Supplementary Material]

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

# A   Concentration Bounds

We state the Chernoff bound and Hoeffding's inequality, the two standard concentration bounds used to bound the error of the estimates of the marginal contributions.

**Lemma 13** (Chernoff Bound). *Let $X_1, \ldots, X_n$ be independent indicator random variables, $X = \sum_{i=1}^{n} X_i$ and $\mu = \mathbb{E}[X]$. For $0 < \delta < 1$,*

$$\Pr[|X - \mu| \geq \delta\mu] \leq 2e^{-\mu\delta^2/3}.$$

**Lemma 14** (Hoeffding's inequality). *Let $X_1, \ldots, X_n$ be independent random variables with values in $[0, b]$. Let $X = \frac{1}{m}\sum_{i=1}^{m} X_i$ and $\mu = \mathbb{E}[X]$. Then for every $0 < \epsilon < 1$,*

$$\Pr\left[|X - \mathbb{E}[X]| \geq \epsilon\right] \leq 2e^{-2m\epsilon^2/b^2}.$$

# B   Estimates of Expected Marginal Contributions

Recall that $S \sim \mathcal{D}_a$ is the set obtained from $S \sim \mathcal{D}$ conditioned on $a \in S$. Similarly, we defined $S \sim \mathcal{D}_{-a}$ by conditioning on $S$ not containing $a$ and we also extended this definition for multiple nodes, such as with $\mathcal{D}_{a,-b}$. Similarly as with $\mathcal{D}_a$ and $\mathcal{D}_{-a}$, let $\mathcal{S}_a$ and $\mathcal{S}_{-a}$ be the collections of samples containing and not containing $a$ respectively. We also extend this notation for multiple nodes, e.g., $\mathcal{S}_{a,-b}$.

The first order marginal contributions $v(a)$ are estimated with the difference $\tilde{v}(a)$ between the average value of samples containing $a$ and the average value of samples not containing $a$, i.e.

$$\tilde{v}(a) := \left( \frac{1}{|\mathcal{S}_a|} \sum_{S \in \mathcal{S}_a} f(S) - \frac{1}{|\mathcal{S}_{-a}|} \sum_{S \in \mathcal{S}_{-a}} |f(S)| \right).$$

Similarly, the second order marginal contribution $v_b(a)$ are estimated with

$$\tilde{v}_b(a) := \left( \frac{1}{|\mathcal{S}_{a,b}|} \sum_{S \in \mathcal{S}_{a,b}} f(S) - \frac{1}{|\mathcal{S}_{-a,b}|} \sum_{S \in \mathcal{S}_{-a,b}} |f(S)| \right).$$

We show that these estimates $\tilde{v}(a)$ and $\tilde{v}_b(a)$ are arbitrarily good when the distribution $\mathcal{D}$ is a bounded product distribution.

**Lemma 15.** *The estimates $\tilde{v}(a)$ and $\tilde{v}_b(a)$ are arbitrarily close to the first and second order marginal contributions $v(a)$ and $v_b(a)$ of a node $a$ given samples from a bounded product distribution $\mathcal{D}$.[4] For all $a, b \in N$ and given $\text{poly}(n, 1/\delta, 1/\epsilon)$ i.i.d. samples from $\mathcal{D}$,*

$$|\tilde{v}(a) - v(a)| \leq \epsilon \quad and \quad |\tilde{v}_b(a) - v_b(a)| \leq \epsilon$$

*with probability at least $1 - \delta$ for any $\delta > 0$.*

*Proof.* We give the analysis for the second order marginal contribution. The proof follows identically for the first order marginal contribution by treating $b$ as null. Note that since $\mathcal{D}$ is a product distribution,

$$\mathbb{E}_{S \sim \mathcal{D}: b \in S, a \notin S}[f(S \cup a)] - \mathbb{E}_{S \sim \mathcal{D}: b \in S, a \notin S}[f(S)] = \mathbb{E}_{S \sim \mathcal{D}: b \in S, a \in S}[f(S)] - \mathbb{E}_{S \sim \mathcal{D}: b \in S, a \notin S}[f(S)].$$

Since marginal probabilities of the product distributions are assumed to be bounded from below and above by $1/\text{poly}(n)$ and $1 - 1/\text{poly}(n)$ respectively, $|\mathcal{S}_{a,b}| = m/\text{poly}(n)$ and $|\mathcal{S}_{-a,b}| = m/\text{poly}(n)$ for all $a$ by Chernoff bound. In addition, $\max_S f(S)$ is assumed to be bounded by $\text{poly}(n)$. So by Hoeffding's inequality,

$$\Pr\left( \left| \frac{1}{|\mathcal{S}_{a,b}|} \sum_{S \in \mathcal{S}_{a,b}} f(S) - \mathbb{E}_{S \sim \mathcal{D}|b \in S, a \in S}[f(S)] \right| \geq \epsilon/2 \right) \leq 2e^{-\frac{m\epsilon^2}{\text{poly}(n)}},$$

for $0 < \epsilon < 2$

$$\Pr\left(\left|\frac{1}{|\mathcal{S}_{-a,b}|}\sum_{S\in\mathcal{S}_{-a,b}}|f(S)| - \mathop{\mathbb{E}}_{S\sim\mathcal{D}|b\in S,a\notin S}[f(S)]\right| \geq \epsilon/2\right) \leq 2e^{-\frac{m\epsilon^2}{\text{poly}(n)}}$$

for $0 < \epsilon < 2$. Thus,

$$\Pr\left(\left|\mathop{\mathbb{E}}_{S\sim\mathcal{D}|b\in S,a\notin S}[f(S\cup a)-f(S)] - \left(\frac{1}{|\mathcal{S}_{a,b}|}\sum_{S\in\mathcal{S}_{a,b}}f(S) - \frac{1}{|\mathcal{S}_{-a,b}|}\sum_{S\in\mathcal{S}_{-a,b}}|f(S)|\right)\right| \geq \epsilon\right)$$
$$\leq 4e^{-\frac{m\epsilon^2}{\text{poly}(n)}}$$

for $0 < \epsilon < 2$.

$\square$

# C   Erdős–Rényi Random Graphs

A $G_{n,p}$ Erdős–Rényi graph is a random graph over $n$ vertices where every edge realizes with probability $p$. Note that the graph obtained by the two step process which consists of first the stochastic block model and then the independent cascade model is a union of $G_{|C|,p_C}$ for each community $C$. The following are seminal results from Erdős–Rényi characterizes phase transitions for $G_{n,p}$ graphs.

**Lemma 3.** *[ER60] Assume $C$ is a "dense" community, then the subgraph $G[C]$ of $G$ is connected with probability $1 - O(|C|^{-2})$.*

*Proof.* Assume $p_C = c\log|C|/|C|$ for $c > 1$. From Theorem 4.6 in [BHK] which presents the result from [ER60], the expected number of isolated vertices $a$ in $G(|C|, p)$ is

$$\mathbb{E}[i] = |C|^{1-c} + o(1)$$

and from Theorem 4.15 in [BHK], the expected number of components of size between 2 and $|C|/2$ is $O(|C|^{1-2c})$. Thus the expected number of components of size at most $n/2$ is $O(n^{1-c})$ and the probability that the graph is connected is $1 - O(|C|^{1-c})$. Finally, since $c \geq 3$ for dense communities, the probability that the graph for community $C$ is connected is $1 - O(|C|^{-2})$.     $\square$

**Lemma 7.** *[ER60] Let $C$ be a tight community with $|C| = \omega(1)$, then $G[C]$ has a "giant" connected component containing a constant fraction of the nodes in $C$ w.p. $1 - o(1)$.*

# D   Missing Analysis from Section 3

**Theorem 6.** *In the dense communities and small seed set setting, COPS with $\alpha$-overlap allowed, for any constant $\alpha \in (0, 1)$ is a $1 - o(1)$-approximation algorithm for learning to influence from samples from a bounded product distribution $\mathcal{D}$.*

*Proof.* First, we claim that a node $a \in C$ is not removed from the ordering if there is no other node from $C$ before $a$. For $b \notin C$, we have

$$v(a) = \mathbb{E}_{S\sim\mathcal{D}_{-a}}[f_S(a)] = \mathbb{E}_{S\sim\mathcal{D}_{-a}}[f_S(a) : b \in S] = \mathbb{E}_{S\sim\mathcal{D}_{-a,b}}[f_S(a)] = v_b(a)$$

where the second equality is since $a$ and $b$ are in different communities and since $\mathcal{D}$ is a product distribution. Thus, $\text{OVERLAP}(a, b, \alpha) = $ False for any $\alpha \in (0, 1]$.

Next, recall that $v(a) \leq |C|$ for all $a \in C$. Thus, by Lemmas 4 and 5. COPS returns a set that contains one node from $k$ different communities that have sizes that are at most a factor $1 - o(1)$ away from the sizes of the $k$ largest communities. Since the $k$ largest communities are connected with high probability, the optimal solution contains one node from each of the $k$ largest communities. Thus, we obtain a $1 - o(1)$ approximation.     $\square$

# E  Missing Analysis from Section 4

**Lemma 8.** *Let $C$ be a loose community, then $f(\{a\}) \leq c$ for all $a \in C$ and some constant $c$.*

*Proof.* Fix a node $a \in C$. We consider a Galton-Watson branching process starting at individual $a$ where the number of offsprings of an individual is $X = \text{Binomial}(|C| - 1, p_C)$. We show that the expected total size $s$ of this branching process is $1/(1 - p_C \cdot (|C| - 1))$ and that this expected size $s$ upper bounds $f(\{a\})$.

We first argue that $s \geq f(\{a\})$. The expected number of nodes influenced by $a$ can be counted via a breadth first search (BFS) of community $C$ starting at $a$. The number of edges leaving a node in this BFS is $\text{Binomial}(|C| - 1, p_C)$, which is exactly the number of offsprings of an individual in the branching process. Since the nodes explored in the BFS are only the nodes not yet explored, the number of nodes explored by BFS is upper bounded by the branching process and we get $\mathbb{E}[s] \geq f(\{a\})$.

Next, let $\mu = p_C \cdot (|C| - 1) < 1$ be the expected number of offsprings of an individual in the branching process. Let $s_i$ be the expected number of individuals at generation $i$ of the branching process. We show by induction that $\mathbb{E}[s_i] = \mu^i$. The base case is trivial for $i = 1$. Next, for $i = 2$,

$$\mathbb{E}[s_i] = \sum_{j=0}^{\infty} \Pr[s_{i-1} = j] \cdot \mathbb{E}[s_i | s_{i-1} = j] = \sum_{j=0}^{\infty} \Pr[s_{i-1} = j] \cdot j \cdot \mu = \mu \,\mathbb{E}[s_{i-1}] = \mu^i$$

where the last inequality is by the inductive hypothesis. Thus, $\mathbb{E}[s] = \sum_{i=0}^{\infty} \mu^i = 1/(1 - \mu)$ since $\mu < 1$. Finally, since $C$ is tight, $\mu \leq 1 - \epsilon$ for some constant $\epsilon > 0$ and $f(\{a\}) \leq \mathbb{E}[s] \leq 1/\epsilon$.  $\square$

**Lemma 11.** *If $a, b \in C$ and $C$ is a tight community, then $\text{OVERLAP}(a, b, \alpha) = \text{True}$ for $\alpha = o(1)$.*

*Proof.* Let $a, b \in C$ s.t. $C$ is a tight community. The marginal contribution of node $a_i$ can be decomposed into the whether $a \in \text{gc}(C)$:

$$v(a) = \Pr_G[a \in \text{gc}(C)] \cdot \mathop{\mathbb{E}}_{S \sim \mathcal{D}_{-a}}[f_S(a) : a \in \text{gc}(C)] + \Pr_G[a \notin \text{gc}(C)] \cdot \mathop{\mathbb{E}}_{S \sim \mathcal{D}_{-a}}[f_S(a) : a \notin \text{gc}(C)]$$

Since $\mathcal{D}$ is a product distribution,

$$\mathop{\mathbb{E}}_{S \sim \mathcal{D}_{-a}}[f_S(a) : a \in \text{gc}(C)] = (\Pr[b \notin \text{gc}(C)] + \Pr_{S \sim \mathcal{D}}[b \in \text{gc}(C), b \notin S]) \mathop{\mathbb{E}}_{S \sim \mathcal{D}_{-a,-b}}[f_S(a) : a \in \text{gc}(C)]$$

$$\geq (1 + \epsilon) \Pr[b \notin \text{gc}(C)] \mathop{\mathbb{E}}_{S \sim \mathcal{D}_{-a,-b}}[f_S(a) : a \in \text{gc}(C)]$$

$$= (1 + \epsilon) \mathop{\mathbb{E}}_{S \sim \mathcal{D}_{-a,b}}[f_S(a) : a \in \text{gc}(C)]$$

for some constant $\epsilon > 0$ since $\Pr_{S \sim \mathcal{D}_{-a}}[b \in \text{gc}(C), b \notin S] \geq \epsilon_1$ for some constant $\epsilon_1 > 0$. Since,

$$\Pr_G[a \in \text{gc}(C)] \mathop{\mathbb{E}}_{S \sim \mathcal{D}_{-a,b}}[f_S(a) : a \in \text{gc}(C)] \geq \epsilon'|C| \geq \epsilon' \Pr_G[a \notin \text{gc}(C)] \cdot \mathop{\mathbb{E}}_{S \sim \mathcal{D}_{-a}}[f_S(a) : a \notin \text{gc}(C)]$$

for some constant $\epsilon' > 0$, we get

$$v(a) \geq (1 + \epsilon) \Pr_G[a \in \text{gc}(C)] \cdot \mathop{\mathbb{E}}_{S \sim \mathcal{D}_{-a,b}}[f_S(a) : a \in \text{gc}(C)]$$

$$+ \Pr_G[a \notin \text{gc}(C)] \cdot \mathop{\mathbb{E}}_{S \sim \mathcal{D}_{-a}}[f_S(a) : a \notin \text{gc}(C)]$$

$$\geq (1 + \epsilon\epsilon'/2) \Big( \Pr_G[a \in \text{gc}(C)] \mathop{\mathbb{E}}_{S \sim \mathcal{D}_{-a,b}}[f_S(a) : a \in \text{gc}(C)]$$

$$+ \Pr_G[a \notin \text{gc}(C)] \cdot \mathop{\mathbb{E}}_{S \sim \mathcal{D}_{-a,b}}[f_S(a) : a \notin \text{gc}(C)] \Big)$$

$$= (1 + \epsilon\epsilon'/2)v_b(a)$$

Thus, $\text{OVERLAP}(a, b, \alpha) = \text{True}$ for $\alpha = o(1)$.  $\square$

**Theorem 12.** *With overlap allowed $\alpha = 1/\text{poly}(n)$, COPS is a constant factor approximation algorithm for learning to influence from samples drawn from a bounded product distribution $\mathcal{D}$ in the setting with tight and loose communities and non-ubiquitous seed sets.*

Figure 3: Performance of COPS as a function of the overlap $\alpha$ allowed. The performance is normalized so that the performance of GREEDY and MARGI corresponds to value 1 and 0 respectively

*Proof.* First, observe that $f(S) \geq k$ since nodes trivially influence themselves. Let $a_i$ be the node picked by the algorithm that is in the $i$th position of the ordering after the pruning and assume $i \leq t$. By Lemma 10, $f(\{a_i\}) \geq \epsilon |C_i|$ where $C_i$ is the $i$th largest tight community with super-constant individual influence and with at least one node in $S^\star$. Thus $a_i$, $i \in [t]$ is in a tight community, otherwise it would have constant influence by Lemma 8, which is a contradiction with $f(\{a_i\}) <\geq \epsilon |C_i|$. Since $a_i$, $i \in [t]$ is in a tight community, by Lemma 11, we obtain that $a_1, \ldots, a_i$ are all in different communities. We denote by $S_t$ the subset of the solution returned by COPS and obtain We obtain $f(S^\star) \leq \sum_{i=1}^{t} |C_i| + c \cdot k \leq \sum_{i=1}^{t} \frac{1}{\epsilon} \cdot f(\{a_i\}) + c \cdot f(S) = \frac{1}{\epsilon} \cdot f(S_t) + c \cdot f(S) \leq c_1 \cdot f(S)$ for some constant $\epsilon, c, c_1$ by Lemmas 9, 10, and since $a_i, a_j$ are in different communities for $i, j \leq t$. $\square$

# F   Additional Experimental Setup and Analysis

**Additional description for the experimental setup.**   Each point in a plot corresponds to the average performance of the algorithms over 10 trials. The default values for $k$ is $k = 10$. For the experiments on synthetic data, the default overlap allowed is $\alpha = 0.5$, for the Facebook experiments $\alpha = 0.4$ and for the DBLP experiments $\alpha = 0.2$. The default edge weights are chosen so that in the random realization of $G$ the average degree of the nodes is approximately 1.

**Additional analysis.**   As discussed in Section 5, the performance of COPS as a function of the overlap allowed (Figure 3) can be explained as follows: Its performance slowly increases as the the overlap allowed increases and COPS can pick from a larger collection of nodes until it drops when it allows too much overlap and picks mostly very close nodes from the same largest community.