[Reviews · NeurIPS 2017]

Reviewer 1



This paper studies the problem of influence maximization in social networks. In the influence process, one selects a set of initially influenced seed nodes in the network, and the influence propagates following a random propagation process, and at the end the expected number of influenced nodes is the influence of the seed set. Influence maximization is then the task of selecting a seed set of a certain size to maximize the influence of the seed set, given samples from the influence process in the form of (a random seed set, the influence of the seed set in an instance of the random propagation process). The paper considers the classical independent cascade model for propagation, and assumes the sample seed set is from a product distribution. The paper proposes an algorithm that enjoys provable guarantees in certain random graph models that can capture the community structure in social networks, thus enjoys good performance in the experiments on both synthetic and real datasets. The idea is to avoid selecting nodes from the same community, since their potential influenced nodes can overlap a lot. This is detected by estimating the second order marginal contribution of the first node w.r.t. the second node. The authors provided performance guarantees under two settings. In the dense communities and small seed set setting, the approximation ratio is 1-o(1), while it is a constant in the setting with tight and loose communities and non-ubiquitous seed sets. The presentation is clear and easy to follow. minor: --missing reference in Line 110, and the title of Figure 1

Reviewer 2



This work marries influence maximization (IM) with recent work on submodular optimization from samples. The work salvages some positive results from the wreckage of previous impossibility results on IM from samples, by showing that under an SBM model of community structure in graphs, positive results for IM under sampling are possible with a new algorithm (COPS) that is a new variation on other greedy algorithms for IM. It’s surprising that the removal step in the COPS algorithm is sufficient from producing the improvement seen between Margl and COPS in Figure 2 (where Margl sometimes does worse than random). Overall this is a strong contribution to the IM literature. Pros: - Brings IM closer to practical contexts by studying IM under learned influence functions - Gives rigorous analysis of this problem for SBMs - Despite simplicity of SBMs, solid evaluation shows good performance on real data Cons: - The paper is very well written, but sometimes feels like it oversimplifies the literature in the service of somewhat overstating the importance of the paper. But it’s good work. - The paper could use a conclusion. 1) The use of “training data” in the title makes it a bit unclear what the paper is about. I actually thought the paper was going to be about something very different, some curious application of IM to understand how training data influences machine learning models in graph datasets. Having read the abstract it made more sense, but the word “training” is only used once beyond the abstract, in a subsection title. It seemed as though the focus on SBMs and community structure deserve mentioning in the title? 2) The SBM is a very simplistic model of “communities,” and it’s important to stay humble about what’s really going on in the theoretical results: the positive results for SBMs in this work hinge on the fact that within communities, SBM graphs are just ER graphs, and the simple structure of these ER graphs is what is powering the results. 3) With (2) registered as a minor grievance, the empirical results are really quite striking, given that the theoretical analysis assumes ER structure within the communities. It’s also notable that the COPS algorithm really does quite well even graphs from on the BA model, which is far from an SBM. In some sense, the issue in (2) sets up an expectation (at least for me) that COPS won’t "actually work". This is quite intriguing, and suggests that the parts of the analysis relying on the ER structure are not using the strongest properties of it (i.e., something close to the main result holds for models much more realistic than ER). This is a strong property of the work, inviting future exploration. Minor points: 1) The dichotomous decomposition of the IM literature is a little simplistic, and some additional research directions deserve mention. There has been several recent works that consider IM beyond submwdular cases, notably the “robust IM” work by [HK16] (which this submission cites as focusing on submwdular IM, not wrong) and also [Angell-Schoenebeck, arxiv 2016]. 2) Two broken cites in Section 3.1. Another on line 292. And I think I see at least one more. Check cites. 3) The name COPS changes to Greedy-Prune in Section 5 (this reviewer assumes they refer to the same thing). 4) I really prefer for papers to have conclusions, even if the short NIPS format makes that a tight fit.

Reviewer 3



The paper proposes a new influence maximization algorithm for the discrete-time independent cascade diffusion model. It also provides a formal approximation analysis on the stochastic block model adjustifying the intuition of the algorithm to exploit the community structure of the underlying networks. Experiments on synthetic and real datasets show that the performance of the new algorithm is close to the greedy upper bound achieved accordingly. The major problem of the paper is that although the analysis of the algorithm assumes a stochastic block model (SBM), in practice, the algorithm unfortunately does not utilize the insights learned from the SBM (since SBM can be fitted in the training data) As a result, it should compare with other heuristics based on community structures. To name a few, "Maximizing Influence Propagation in Networks with Community Structure"; "Correction: A Novel Top-k Strategy for Influence Maximization in Complex Networks with Community Structure"; "A New Community-based Algorithm for Influence Maximization in Social Network"; Furthermore, on networks with known community structures, it should at least compare with one basic heuristic "Scalable influence maximization for independent cascade model in large-scale social networks" demonstrating that the new algorithm perform better compared to alternatives which do not exploit the community structures.